# Sediment transport in South Asian rivers high enough to impact satellite gravimetry

Alexandra Klemme[1], Thorsten Warneke[1], Heinrich Bovensmann[1], Matthias Weigelt[2], Jürgen Müller[2], Tim Rixen[3], Justus Notholt[1], and Claus Lämmerzahl[4]

[1]Institute of Environmental Physics, University of Bremen, Otto-Hahn-Allee 1, 28359 Bremen, Germany
[2]Institute of Geodesy, Leibniz Universität Hannover, Schneiderberg 50, 30167 Hannover, Germany
[3]Leibniz Center for Tropical Marine Research, Fahrenheitstr. 6, 28359 Bremen, Germany
[4]Centre of Applied Space Technology and Microgravity, University of Bremen, Am Fallturm 2, 28359 Bremen, Germany

**Correspondence:** A. Klemme (aklemme@uni-bremen.de)

**Abstract.** Satellite gravimetry is used to study the global hydrological cycle. It is a key component in the investigation of groundwater depletion on the Indian subcontinent. Terrestrial mass loss caused by river sediment transport is assumed to be below the detection limit in current gravimetric satellites of the Gravity Recovery and Climate Experiment Follow-On mission. Thus, it is not considered in the calculation of terrestrial water storage (TWS) from such satellite data. However, the Ganges and Brahmaputra rivers, which drain the Indian subcontinent, constitute one of the world's most sediment rich river systems. In this study, we estimate the impact of sediment mass loss within their catchments on local trends in gravity and consequential estimates of TWS trends. We find that for the Ganges-Brahmaputra-Meghna catchment, sediment transport accounts for $(4 \pm 2)\%$ of the gravity decrease currently attributed to groundwater depletion. The sediment is mainly eroded from the Himalayas, where correction for sediment mass loss reduces the decrease in TWS by $0.22\,\mathrm{cm}$ of equivalent water height per year $(14\%)$. However, sediment mass loss in the Brahmaputra catchment is more than twice that in the Ganges catchment, and sediment is mainly eroded from mountain regions. Thus, the impact on gravimetric TWS trends within the Indo-Gangetic plain - the main region identified for groundwater depletion - results to be comparatively small $(< 2\%)$.

## 1 Introduction

Since March 2002, the Gravity recovery and Climate Experiment (GRACE) provides satellite based measurements of the Earth's gravity field (Dahle et al., 2019), with the only major data gap being between the end of the original satellite mission in August 2017 and the launch of the follow-on mission (GRACE-FO) in May 2018. Gravity fields derived from satellite measurements yield information on global mass variations, which have proven crucial to monitor changes in global water storage and fluxes (Rodell et al., 2018). Retrieved data of the mass equivalent water height (EWH) are widely used for studies on topics such as glacier melting (Jacob et al., 2012; Luthcke et al., 2013), groundwater depletion (Rodell et al., 2009; Xie et al., 2020) and sea level rise (Cazenave et al., 2009; Jeon et al., 2018).

One significant region that yields negative trends in terrestrial water storage (TWS) is north-west India with an average decrease of $(29 \pm 2.5)\,\mathrm{m^3\,H_2O\,yr^{-1}}$ (Rodell et al., 2018; Xie et al., 2020). Several studies have investigated this decrease

and explained it by a large-scale groundwater loss due to excessive extraction for irrigation (Tiwari et al., 2009; Rodell et al., 2009; Panda and Wahr, 2016; Rodell et al., 2018; Xie et al., 2020). Wada et al. (2012) found that the use of non-renewable groundwater for irrigation more than trippled since 1960. In the year 2000, one-fifth of the global irrigation water demand was fed by non-renewable groundwater abstraction, with the majority being abstracted in India and Pakistan (Wada et al., 2012). Furthermore, the depletion in Indian groundwater occurred during a period of increased precipitation, implying an even stronger water deficit for future droughts (Rodell et al., 2018).

A large fraction of the Indian subcontinent is drained by the Ganges-Brahmaputra river system. The Ganges and Brahmaputra rivers originate in the Himalayan belt and drain intensely cultivated regions before their confluence in Bangladesh and discharge into the Bay of Bengal (Subramanian and Ramanathan, 1996; Garzanti et al., 2011). These rivers are one of the largest source of water and sediment to the world's ocean (Akter et al., 2021). The high amounts of sediment they carry into the Bay of Bengal make up the Bengal Delta and Submarine Fan that extends from Bangladesh to south of the equator and contains at least $1.1 \cdot 10^{19}\,\mathrm{kg}$ of sediment with an average accumulation rate of $665 \cdot 10^9\,\mathrm{kg\,yr^{-1}}$ (Curray, 1994). The sediment transport by the Ganges-Brahmaputra river system shows strong diurnal, seasonal, and inter-annual variations (Subramanian and Ramanathan, 1996). Estimates of sediment discharge vary widely between $200 \cdot 10^9\,\mathrm{kg\,yr^{-1}}$ and $1,600 \cdot 10^9\,\mathrm{kg\,yr^{-1}}$ for the Ganges River (Rahman et al., 2018; Holeman, 1968) and between $150 \cdot 10^9\,\mathrm{kg\,yr^{-1}}$ and $1,157 \cdot 10^9\,\mathrm{kg\,yr^{-1}}$ for the Brahmaputra River (Akter et al., 2021; Milliman and Meade, 1983). Yet, recent studies state the annual combined sediment discharge of the rivers to be about $10^{12}\,\mathrm{kg}$ with the majority being carried during the monsoon season from June to October (Wasson, 2003; Kuehl et al., 2005; Wilson and Goodbred, 2015; Mouyen et al., 2018; Mahmud et al., 2020; Akter et al., 2021).

This river sediment transport implies a terrestrial mass reduction that has so far not been considered in the computation of gravimetric TWS data. A study by Schnitzer et al. (2013) found that the mass loss associated with the large-scale soil erosion in the Chinese Loess Plateau was not visible considering the available GRACE resolution. However, recent studies found the sediment discharge to the ocean to be visible using satellite gravimetry of the estuary regions (Mouyen et al., 2018; Li et al., 2022). While the incorporation of sediment mass loss into monthly GRACE solutions over land might be impossible at the current satellite resolutions, it is a non-negligible loss when considering long term TWS trends studied in regard to e.g. groundwater depletion.

Additional processes to consider in long-term gravimetric data are plate tectonics. The Himalaya mountain range experiences uplift due to the tectonic collision between the Indian and the Eurasian continental plates. The gravimetric impact of this process is not the focus of this study. Yet, knowledge of such additional tectonic process is essential to contextualize the resulting sediment impact, as the increase in mass due to this Himalayan mountain uplift could counteract part of the mass loss due to sediment erosion and discharge.

In this study, we estimate this impact of mass loss due to soil erosion and sediment transport by major rivers draining the Indian subcontinent on TWS trends observed by the GRACE and GRACE-FO satellites.

## 2  Methods

### 2.1  Study Area

This study focuses on the Ganges and Brahmaputra catchments, with some discussion of the Indus and Meghna catchments. The rivers are located mainly in Northern India but also partly flow through China, Pakistan, Nepal, Bhutan, Afghanistan and Bangladesh (Figure 1). The river catchments are impacted by the South Asian monsoon, bringing high precipitation and river discharge from June to October. The Ganges and Brahmaputra rivers originate in the Himalayan mountain belt and discharge into the Bay of Bengal after confluence with the Meghna river in Bangladesh. Together with the Indus River, they drain the majority of the Himalayas.

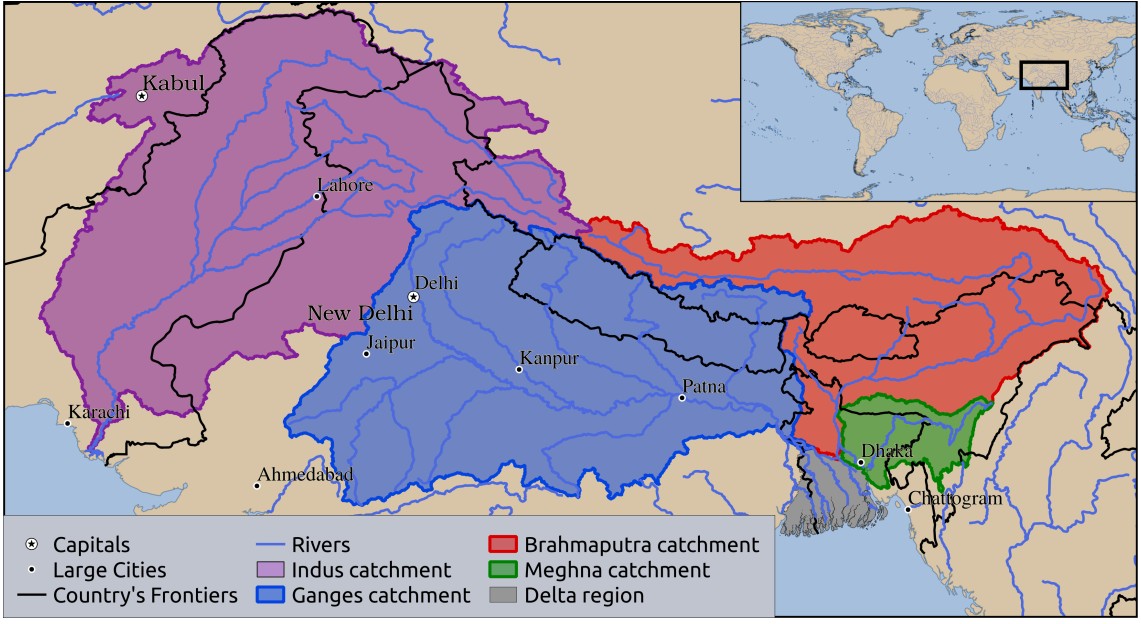

**Figure 1.** Map of investigated catchments (Lehner and Grill, 2013) and river paths (GRDC, 2020).

Due to high erosion rates in the Himalayan mountain region, sediment concentrations in these rivers are among the highest worldwide (Subramanian and Ramanathan, 1996; Akter et al., 2021). Especially the Brahmaputra catchment has a large mountain fraction, while the other river catchments show higher agricultural fractions (Table 1). A map including the locations of mountain ranges and agricultural land as well as more detailed river descriptions are included in the supplemental material.

India hosts the world's largest groundwater-reliant agricultural irrigation system (Xie et al., 2020). Of its total irrigation-equipped area ($620,000\,\mathrm{km}^2$), about $64\%$ can be irrigated with groundwater, amounting to a total consumptive groundwater use for irrigation of about $200\,\mathrm{km}^3\,\mathrm{yr}^{-1}$ (Siebert et al., 2010). The fraction of irrigation reliant on groundwater has increased over the past decades from only $29\%$ in 1951 to more than $50\%$ in 2022 (FAO, 2022), with the absolute groundwater irrigated

area being more than 5 times larger than in 1951 (Siebert et al., 2010; FAO, 2022). The major groundwater aquifer for the studied regions is located in the Indo-Gangetic Plain and stretches mainly beneath the Indus and Ganges floodplains, while there are only shallow aquifers in the Himalayan mountain regions (supplemental Figure S2).

**Table 1.** Mountain and agricultural fractions of the catchments.

|  | Total | GBM | Ganges | Brahmaputra | Meghna | Indus |
|---|---|---|---|---|---|---|
| catchment area ($km^2$) | 2,679,069 | 1,576,134 | 950,754 | 539,989 | 85,391 | 1,102,935 |
| mountain fraction (%) | 36.0 | 32.9 | 15.9 | 67.4 | 3.3 | 51.6 |
| agricultural fraction (%) | 45.6 | 39.3 | 65.2 | 18.2 | 42.8 | 34.4 |

Total refers to the combined Ganges, Brahmaputra, Meghna and Indus catchments. GBM is the Ganges-Brahmaputra-Meghna catchment. Mountain fraction refers to regions of elevation $\geq 1,500\,m$ (based on elevation data from Jarvis et al., 2008). Agricultural regions are from GLCNMO (2017). River catchment data are from Lehner and Grill (2013).

## 2.2 Gravimetry and sediment data

Gravimetry data in this study is from the GRACE and GRACE-FO satellites. We use post-processed data from the Combination International Service for Time-variable Gravity Fields (COST-G) Level 3 data product (Boergens et al., 2020) for TWS anomalies in units of EWH. The data are based on the COST-G RL01 Level 2B products by Dahle and Murböck (2020) and include gridded data for TWS, TWS uncertainty, spatial leakage contained in the TWS and the background model atmospheric mass, all in a monthly resolution of $1° \times 1°$. The potential impact of filtering and spatial leakage in these data is discussed in the Supplemental Material.

Monthly TWS anomalies within the investigated catchments are derived by selecting all data whose grid centers are located within the respective catchment and calculating their area weighed average for each month. Data uncertainty is derived analogously from the area-weighed average of the TWS uncertainties provided in the COST-G data product. Linear least-squares optimizations of the generated monthly time-series yield the local TWS trends. Trend uncertainties contain the standard error of the derived slope optimization as well as the uncertainty of the monthly time series. A more detailed trend analysis is included in the supplemental material.

Sediment data for this study were collected from the literature. Generally, measurements in the study area are scarce and existing data is located close to Bangladesh, providing no information on the areal distribution of sediment loss in the upper catchments. The Supplemental Material provides a discussion on this scarcity in sediment data and the consequences for our study. Complete lists of the sediment data and their sources for the Ganges and Brahmaputra rivers are available in the supplemental tables S1 and S2, respectively.

## 3 Results & Discussions

### 3.1 Geodetic observations of the decrease in terrestrial water storage

Gravimetric data of TWS generally show negative trends within the studied catchments. Trends are most pronounced in the eastern Brahmaputra catchment and in the western Ganges catchment at the border to the Indus catchment. The data yields the strongest decline of $5.8\,\mathrm{cm\,yr^{-1}}$ in north-west India at about 28°N and 76°E (Figure 2).

Comparison of average TWS trends within the individual catchments yield the strongest decrease for the Ganges catchment, followed by the Brahmaputra and Indus catchments. The Meghna catchment shows the weakest trend (Table 2). Low standard deviation of trends in the Brahmaputra and Meghna catchments imply rather homogeneous distributions of the TWS decrease in those catchments (Table 2). Higher standard deviations in the Ganges and Indus catchments (Table 2) are likely caused by the distinct negative trend in north-west India. This is confirmed further by the comparatively low median trend values within these catchments (Table 2).

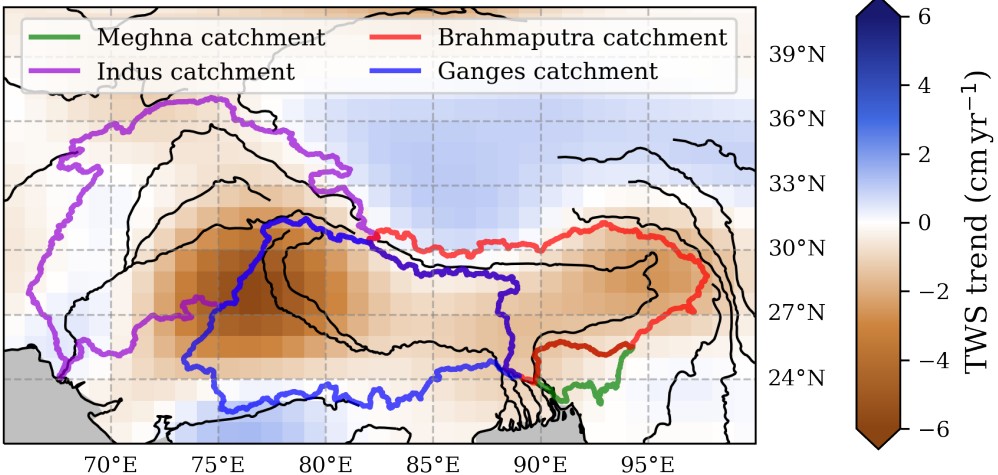

**Figure 2.** Trend of satellite based terrestrial water storage (TWS) with location of major river basins on the Indian subcontinent. Data were derived from linear least-squares approximation of the COST-G data (Boergens et al., 2020), based on the GRACE and GRACE-FO time period of 04-2002 to 12-2022. Location of river catchments are from Lehner and Grill (2013).

Additional assessment of TWS trends in catchment mountain regions yields similar results for the Ganges and the Brahmaputra catchments (Table 2). For the Brahmaputra catchment, the observed TWS decrease is slightly higher than for the catchment average. For the Ganges catchment, it is slightly lower than the catchment average (Table 2). While the center of the main TWS decrease in the Ganges catchment is located in the Indo-Gangetic plain, it extends into the Ganges mountain ranges. This implies that the TWS decrease in the Ganges mountain regions could be overestimated due to the impact of TWS leakage caused by data filtering, as discussed in the Supplemental Material.

**Table 2.** Loss of terrestrial water storage within the catchments.

| TWS loss (cm yr$^{-1}$) | Total | GBM | Ganges | Brahmaputra | Meghna | Indus | Ganges-m | Brahmaputra-m |
|---|---|---|---|---|---|---|---|---|
| mean | 1.35 | 1.51 | 1.63 | 1.45 | 0.60 | 1.13 | 1.56 | 1.60 |
| median | 1.09 | 1.32 | 1.24 | 1.46 | 0.62 | 0.57 | 1.30 | 1.68 |
| standard deviation | 1.43 | 1.36 | 1.67 | 0.64 | 0.35 | 1.49 | 0.71 | 0.66 |
| minimum | -1.12 | -1.12 | -1.12 | 0.27 | 0.09 | -0.48 | 0.94 | 0.28 |
| maximum | 5.78 | 5.77 | 5.77 | 2.64 | 1.17 | 5.78 | 3.40 | 2.64 |

Data show the loss of TWS in cm of equivalent water height per year. Negative values represent a water increase. GBM is the combined
Ganges-Brahmaputra-Meghna catchment. Total refers to the combination of the Ganges, Brahmaputra, Meghna, and Indus catchments. Ganges-m and
Brahmaputra-m refer to the mountain regions (altitude $\geq 1,500$ m) within the Ganges and Brahmaputra catchment, respectively. Data was derived based on
pixel-wise linear least-squares fit of the COST-G GACE data. The mean values are weighed by the different pixel areas while the other statistical variables do
not consider respective pixel sizes.

For the combined study area, the average TWS decrease derived from satellite data is $(1.4 \pm 0.2)\,\mathrm{cm\,yr^{-1}}$. The time series
of TWS in the study area decreases fairly linear with annual variations, mainly driven by precipitation patterns that cause
increasing TWS during the monsoon months and decreasing TWS during dry periods (Figure 3). This TWS decrease over the
complete study area represents a mass reduction of $36 \cdot 10^{12}\,\mathrm{kg\,yr^{-1}}$.

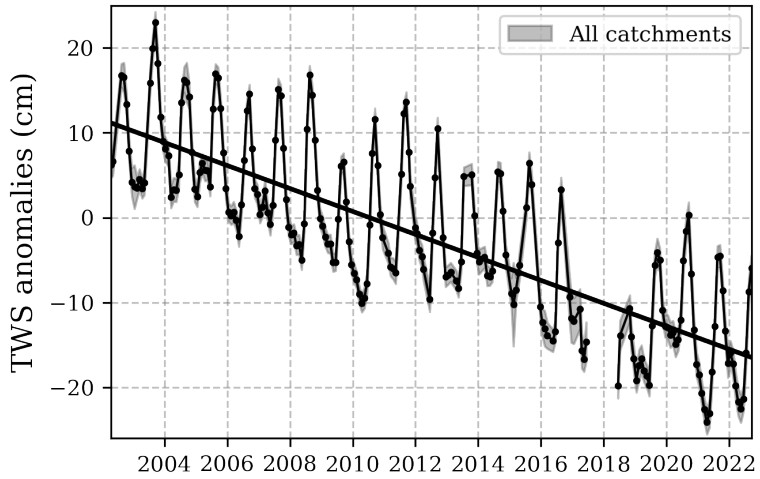

**Figure 3.** Time series of average terrestrial water storage (TWS) anomalies within the combined Ganges, Brahmaputra, Meghna, and Indus
catchments. Data points are area weighed monthly averages within the catchments and shaded areas represents area weighed uncertainties
stated in the COST-G data product (Boergens et al., 2020). The linear trend was derived based on ordinary least-squares optimization of
monthly data. The data gap represents the time between the end of the initial GRACE mission and the start of the GRACE-FO mission.

## 3.2 Mass loss caused by river sediment transport

To estimate the impact of sediment transport on the observed trend in gravity anomalies, we need the total sediment discharge from the studied regions. Based on data collected in various studies, the annual sediment discharge from the Ganges and Brahmaputra rivers is $501 \cdot 10^9 \, \mathrm{kg \, yr^{-1}}$ and $596 \cdot 10^9 \, \mathrm{kg \, yr^{-1}}$, respectively (Table 3). Sediment discharge from the Indus River is $168 \cdot 10^9 \, \mathrm{kg \, yr^{-1}}$ and the Meghna River discharges $11 \cdot 10^9 \, \mathrm{kg}$ of sediment per year (Table 3). The high sediment values in the Ganges and Brahmaputra rivers are caused by their origin in the Himalayan mountains, as those are highly erosion prone regions. The Meghna river originates in the Indian Naga Hills at less than $2,000 \, \mathrm{m}$ elevation and mainly drains the floodplains. The Indus river originates in the Himalayas. However, its annual sediment discharge has been strongly reduced by the installment of dams along the river.

**Table 3.** River sediment transport within the catchments.

| sediment load ($10^9 \, \mathrm{kg \, yr^{-1}}$) | Total | GBM | Ganges | Brahmaputra | Indus | Meghna |
|---|---|---|---|---|---|---|
| mean | 1,276 | 2,008 | 501 | 596 | 168 | 11 |
| median | 1,207 | 1,082 | 480 | 590 | 125 | 12 |
| standard deviation | 633 | 511 | 272 | 237 | 122 | 2 |
| minimum | 400 | 350 | 200 | 150 | 50 | 0 |
| maximum | 3,147 | 2,777 | 1,600 | 1,157 | 370 | 20 |

Sediment loads as compiled from the literature. Total refers to the sum of sediment discharge in all four rivers. GBM refers to sediment discharge in the Ganges-Brahmaputra-Meghna river system. The complete lists of data compiled for the Ganges and Brahmaputra rivers are in the Supplemental Material in Table S2 and Table S3, respectively. Sediment load in the Meghna River is compiled from Coleman (1969), Smith et al. (2009), and Rahman et al. (2018). Sediment load in the Indus River is compiled from Holeman (1968), Milliman and Meade (1983), Giosan et al. (2006), and Mouyen et al. (2018).

Due to data scarcity, it is difficult to assess spatially resolved data for sediment induced gravity changes in the Indian subcontinent. In the following, we separate between sediment eroded from specific mountain regions based on published literature (Wasson, 2003; Galy et al., 2007; Faisal and Hayakawa, 2022). Additionally, a discussion of spatially resolved sediment loss based on soil loss data from the Revised Universal Soil Loss Equation (RUSLE, Borrelli et al., 2017) is included in the Supplemental Material.

The majority of sediment is discharged during the monsoon season from June to October, when there is also high water discharge in the rivers (Islam, 2016). Over the considered period of GRACE measurements (2002-2022), the rivers discharged more than $25 \, \mathrm{Pg}$ of sediment. The average discharge rate is roughly $1.3 \cdot 10^{12} \, \mathrm{kg \, yr^{-1}}$ (Table 3).

## 3.3 Discussion of data seasonality

The seasonality of both TWS anomalies and river sediment discharge depends on the South Asian monsoon. As such, both parameters follow the seasonality of regional precipitation with the sediment discharge peaking approximately one month after the precipitation maximum and the TWS peaking one month after that (Figure 4). Since the monsoon moves from south-east

over the Indian subcontinent, precipitation in the Brahmaputra and Meghna catchments start to increase earlier in the year and
more gradually, while precipitation in the Ganges and Indus catchments start later and increases more rapidly.

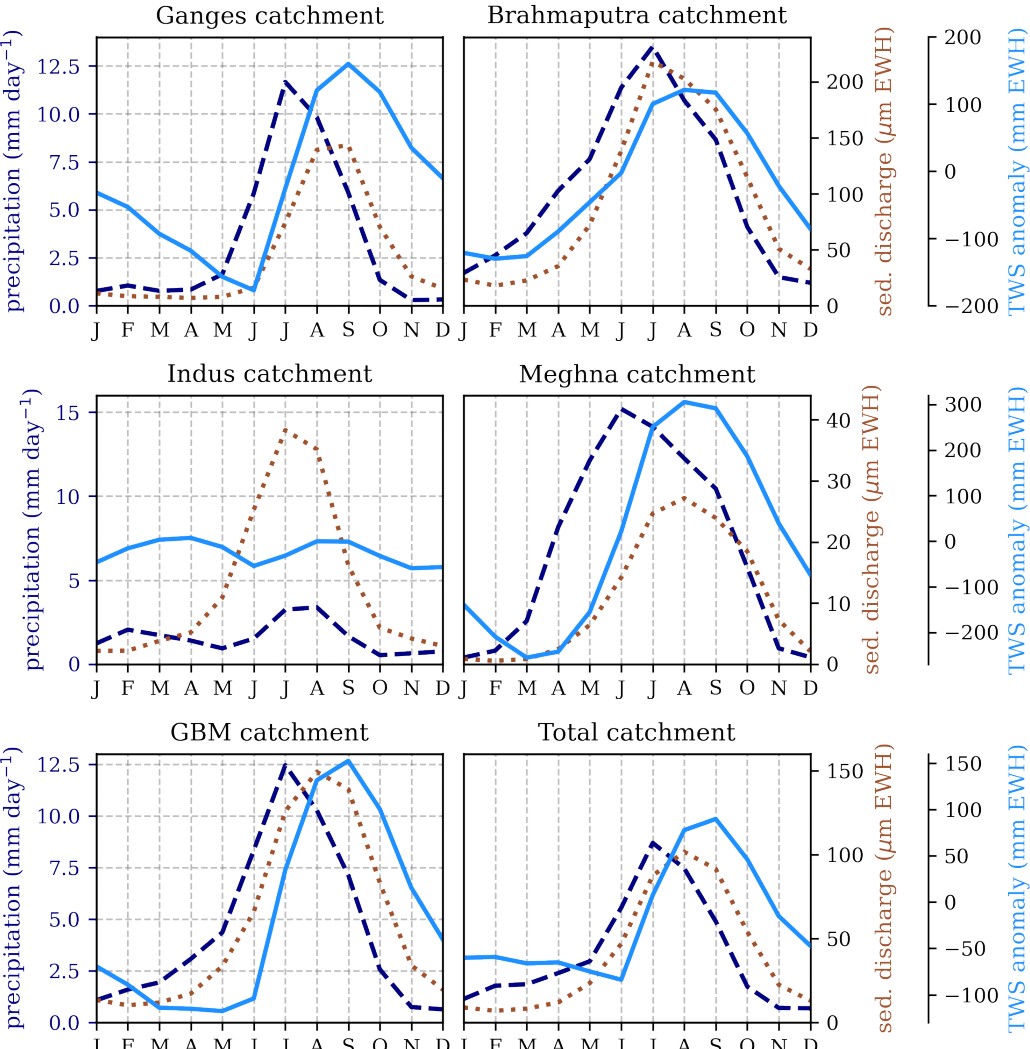

**Figure 4.** Average seasonality of the precipitation (dashed), the terrestrial water storage (TWS, solid), and the sediment discharge (dotted) within the individual Ganges, Brahmaputra, Indus, and Meghna catchments as well as the combined Ganges-Brahmaputra-Meghna catchment (GBM), and the total combined GBM and Indus catchments (Total). Precipitation data are averaged from the ERA5 reanalysis product for 2000-2022 (C3S, 2017). Seasonal TWS anomalies are averaged for the COST-G data product for 2002-2022 Boergens et al. (2020). Seasonality of sediment discharge is based on river water discharge according to data in Islam (2016).

This difference in precipitation patterns is also visible in the sediment discharge and TWS anomalies. For the Brahmaputra River, sediment discharge and TWS in the river catchment yield minima in February and show a gradual increase until the monsoon peak in July (Figure 4). After that, sediment discharge decreases with the precipitation decrease, while TWS stays

high until October, when precipitation rates drop below $5\,\mathrm{mm\,day^{-1}}$. Parameters in the Meghna catchment follow a similar seasonality, whereat precipitation and TWS anomalies are more pronounced in that catchment. Yet, sediment discharge is by an order of magnitude weaker than in the Brahmaputra catchment.

For the Ganges River, sediment discharge increases from June to August and decreases from September to November. TWS anomalies in the Ganges catchment increase between June and August and show a steady decline from September to June, when the precipitation rate is below $6\,\mathrm{mm\,day^{-1}}$ (Figure 4). In the Indus catchment, precipitation rates and TWS anomalies show only small changes during the monsoon season. Additionally, these parameters yield a second local maximum between February and April (Figure 4). This is likely caused by mid-latitude extra-tropical western disturbances in the southern part of the catchment (Cannon et al., 2015). The Indus sediment discharge shows only the one maximum during monsoon season.

Generally, the mass change due to sediment transport reduces gravity values during TWS increase and does not effect gravity observations during TWS decrease. However, the sediment mass loss in units of EWH show values that are by three orders of magnitude smaller than the seasonality observed in GRACE data. This monthly sediment impact is within the uncertainty of monthly gravimetry data and will not considerably impact this study's analysis. While seasonality is included in the following data, we will from here on focus on linear trends in both water and sediment loss.

### 3.4 Impact of sediment transport on geodetic observations of trends in terrestrial water storage

**Table 4.** Sediment impact on gravimetric observations of TWS trends for studied catchments.

| river | catchment area ($\mathrm{km^2}$) | sediment loss ($10^{12}$ kg/yr) | GRACE TWS loss (mm/yr) | abs. sediment impact ($\mathrm{kg/m^2/yr} \approx$ mm/yr) | rel. sediment impact (%) |
|---|---|---|---|---|---|
| Total | $2,679,069$ | $1.28 \pm 0.63$ | $13.5 \pm 2.2$ | $0.48 \pm 0.23$ | $3.6 \pm 2.3$ |
| GBM | $1,576,134$ | $1.11 \pm 0.51$ | $15.1 \pm 2.7$ | $0.70 \pm 0.32$ | $4.6 \pm 3.0$ |
| Ganges | $950,754$ | $0.50 \pm 0.27$ | $16.3 \pm 2.8$ | $0.53 \pm 0.29$ | $3.3 \pm 2.3$ |
| Brahmaputra | $539,989$ | $0.60 \pm 0.24$ | $14.5 \pm 2.6$ | $1.10 \pm 0.44$ | $7.6 \pm 4.4$ |
| Meghna | $85,391$ | $0.011 \pm 0.002$ | $6.0 \pm 4.0$ | $0.13 \pm 0.02$ | $2.2 \pm 1.8$ |
| Indus | $1,102,935$ | $0.17 \pm 0.12$ | $11.3 \pm 1.9$ | $0.15 \pm 0.11$ | $1.3 \pm 1.2$ |
| Ganges-m | $148,948$ | $0.50 \pm 0.27$[b] | $15.6 \pm 2.5$ | $3.36 \pm 1.83$ | $21.5 \pm 15.2$ |
| Ganges-HH | $57,025$ | $0.45 \pm 0.27$ | $15.6 \pm 2.5$[a] | $7.89 \pm 4.74$ | $50.6 \pm 38.6$ |
| Ganges-LH | $91,885$ | $0.05 \pm 0.05$ | $15.6 \pm 2.5$[a] | $0.54 \pm 0.54$ | $3.5 \pm 4.0$ |
| Brahmaputra-m | $361,509$ | $0.60 \pm 0.24$[b] | $16.1 \pm 2.3$ | $1.65 \pm 0.66$ | $10.3 \pm 5.6$ |
| Brahmaputra-NBS | $21,600$ | $0.27 \pm 0.20$ | $16.1 \pm 2.3$[a] | $12.50 \pm 9.26$ | $77.6 \pm 68.6$ |
| Brahmaputra-rem. | $339,900$ | $0.33 \pm 0.22$ | $16.1 \pm 2.3$[a] | $0.97 \pm 0.65$ | $6.0 \pm 4.9$ |

Total refers to the combined Ganges, Brahmaputra, Meghna, and Indus catchments. GBM is the Ganges-Brahmaputra-Meghna catchment. Ganges-m and Barhmaputra-m refer to the mountain regions (altitude $\geq 1,500\,\mathrm{m}$) within the Ganges and Brahmaputra catchment, respectively. Ganges-HH and Ganges-LH refer to the High Himalayas and the Lesser Himalayas in the Ganges catchment, respectively. Brahmaputra-NBS and Brahmaputra-rem. refer to the Namcha Barwa syntaxis and the remaining Brahmaputra mountains, respectively. [a]TWS trends within specific locations in the catchment mountain regions are approximated by the average TWS trend over the mountains. [b]Sediment data for the mountain regions assume all river sediment being eroded from these regions.

### 3.4.1 Impact within the full study area

To compare the mass loss from river sediment transport to the observed TWS trends, the absolute sediment mass loss is divided by the respective catchment area and the density of water. This yields the impact of sediment mass loss in units of EWH. Considering the total catchment size of the Ganges, Brahmaputra, Meghna and Indus rivers (Table 1) as well as their combined sediment discharge (Table 3), this yields an absolute sediment mass impact of roughly $0.5\,\mathrm{mm\,yr^{-1}}$ that is not considered when deriving TWS based on gravimetric observations. Accordingly, this sediment mass loss needs to be subtracted from the

observed trends in TWS anomalies, reducing the local TWS trend of $1.35\,\mathrm{cm\,yr^{-1}}$ by roughly $4\,\%$ (Table 4, Figure 5).

The average monthly sediment impact on TWS observations is less than $0.01\,\mathrm{cm}$ of EWH, which is well within the uncertainties stated for GRACE TWS data in the study area (average $\mathrm{TWS_{std}} \approx 1.4\,\mathrm{cm}$, Boergens et al., 2020). However, considering the whole 20-year time-series, our results imply that a gravity decrease corresponding to $1\,\mathrm{cm}$ EWH currently attributed to groundwater depletion on the Indian subcontinent could be caused by sediment transport instead.

Exclusion of the Indus catchment yields a stronger relative impact of sediment mass loss on the observed TWS trend for the Ganges-Brahmaputra-Meghna catchment. This is caused by higher sediment discharge per catchment area (Table 4). The measured TWS decrease in the Ganges-Brahmaputra-Meghna catchment is slightly higher than for the complete study area (Figure 5). The absolute sediment impact on gravity is $0.7\,\mathrm{kg\,m^{-2}\,yr^{-1}}$. This represents about $4.6\,\%$ of the observed gravity reduction in the Ganges-Brahmaputra-Meghna catchment that is currently attributed to groundwater loss (Table 4). Over the

total GRACE data period, correction for this sediment mass loss would reduce the estimated TWS loss by about $1.6\,\mathrm{cm}$.

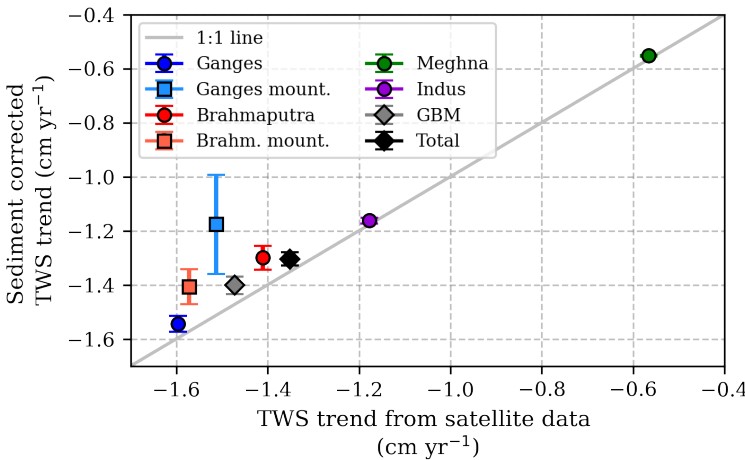

**Figure 5.** Comparison plot between regional trends in terrestrial water storage (TWS) derived from COST-G data product (Boergens et al., 2020) and the trends corrected for sediment mass loss. Data points include the individual catchments as well as the combined Ganges-Brahmaputra-Meghna catchment (GBM), the total combined Indus, Ganges, Brahmaputra, and Meghna catchments (Total) and the mountain fractions of the Ganges (Ganges mount.) and Brahmaputra (Brahm. mount.) catchments. Full time-series of TWS data with and without sediment correction are included in the supplemental figures S16 to S21.

### 3.4.2 Impact within individual catchments

Investigation of the individual river catchments yields the highest sediment mass loss for the Brahmaputra catchment (Table 4). This is consistent with the high fraction of mountains in this catchment (Table 1) and high precipitation rates that enhance erosion in the Eastern Himalayas (Figure 4, Burbank et al., 2012). The absolute sediment mass loss in the Ganges catchment is similar to that in the Brahmaputra catchment (Table 4). However, the Ganges catchment is larger than the Brahmaputra catchment, resulting in a sediment impact per catchment area that is only half that in the Brahmaputra catchment (Table 4). Sediment mass loss in the Meghna and Indus catchments is significantly lower than in the Ganges and Brahmaputra catchments (Table 4).

The Brahmaputra catchment also yields the highest relative impact of sediment mass loss on the observed gravity trend (Table 4). Correction for this impact reduces the TWS decline by $7.8\%$, which over the whole GRACE data period represents more than $2\,\mathrm{cm}$ (Figure 6).In the Ganges catchment, sediment transport represents $3.3\%$ of the gravity decrease, and the impact within the Indus and Meghna catchments is even smaller Figure 5.

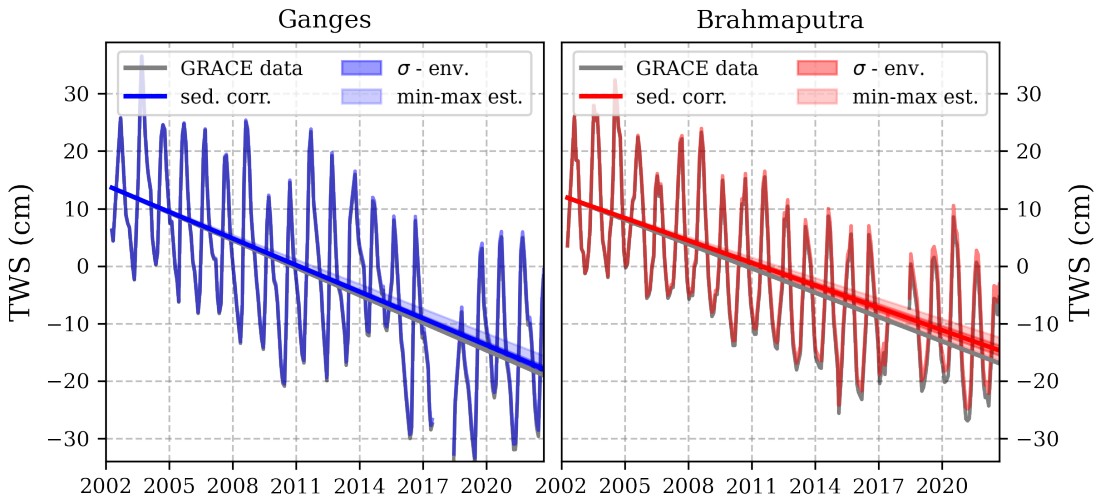

**Figure 6.** Time series of TWS derived from GRACE data (grey) and TWS data corrected for sediment mass loss (color). Data show average over the whole Ganges (left) and Brahmaputra (right) catchments. Ranges for the $\sigma$-environment and the min-max estimates refer to the standard deviation as well as minimum and maximum estimates of sediment discharge as stated in Table 3. Analogue figures for all catchments can be found in the supplemental figures S16 to S21.

### 3.4.3 Impact within the Himalayan mountain regions

Studies agree that the majority of sediment discharged into the Bay of Bengal is derived from the Himalaya mountain ranges (Wasson, 2003; Galy et al., 2007; Faisal and Hayakawa, 2022). Thus, we specifically studied the impact of sediment mass loss in these regions.

The Brahmaputra catchment includes a mountain fraction of $67.4\%$ (Table 1). Assuming all of the river's sediment to be derived from these regions yields a sediment mass loss of $1.7\,\mathrm{kg\,m^{-2}\,yr^{-1}}$ (Table 4). Considering the average TWS decrease derived from GRACE data for the region (Table 4), the sediment mass loss accounts to roughly $10\%$ of the gravity decrease (Figure 7). According to Faisal and Hayakawa (2022), about half $((45\pm15)\%)$ of the Brahmaputra's sediment is derived from the Namcha Barwa syntaxis, the easternmost Himalayan sytaxis that encompasses only $\approx4\%$ of the Brahmaputra catchment. The remaining sediment is derived from Himalayan tributaries that join the Brahmaputra in the Himalayan foreland (Faisal and Hayakawa, 2022). This indicates that local sediment mass loss within the Namcha Barwa syntaxis and the remaining Brahmaputra mountain areas represent $78\%$ and $6\%$ of the observed gravity decrease, respectively (Table 4).

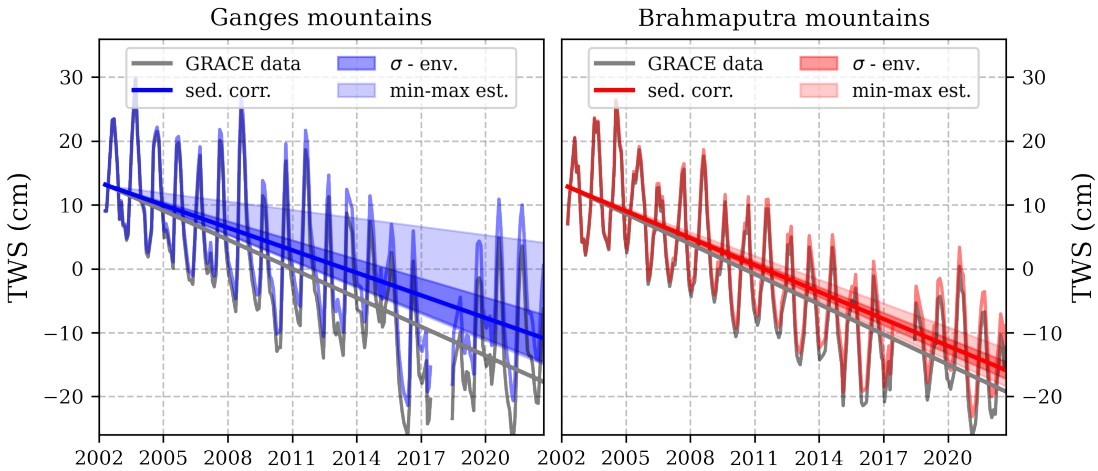

**Figure 7.** Time series of TWS derived from GRACE data (grey) and TWS after the correction for sediment mass loss (color). Data show average over the mountain fraction within the Ganges catchment (left) and the Brahmaputra catchment (right). $\sigma$ environment and min-max estimates refer to the standard deviation as well as minimum and maximum estimates of sediment discharge as stated in Table 3. An analogous figure for the mountain sub-regions is included in the supplement as Figure S22.

The Ganges catchment includes a mountain fraction of only $15.9\%$ (Table 1). Even though sediment discharge in the Ganges river is smaller, the area weighed mass loss over the mountains is about double that of the Brahmaputra mountains (Table 4). Considering the higher TWS decrease in the Ganges mountains, this sediment mass loss accounts for $22\%$ of the gravity decrease observed in the area (Figure 7). According to Faisal and Hayakawa (2022), $(90\pm5)\%$ of the Ganges sediment is derived from the High Himalayas. The remaining sediment is mostly from the Lesser Himalayas (Wasson, 2003) with a smaller contribution from intensely cultivated floodplain regions (Galy et al., 2007; Garzanti et al., 2011). Considering this, the local sediment loss from the High Himalayas represents about half the observed gravity decrease, while in the Lesser Himalayas it is about $4\%$ (Table 4).

### 3.4.4 Impact within floodplain regions

To estimate the impact of sediment discharge on gravity data of groundwater depletion, we are interested in erosion within the Indo-Gangetic floodplain, where the strongest gravity decrease is observed. Generally, the estimation of the sediment impact in river lowlands and floodplains is more complicated due to sedimentary redistribution within the catchments. While some sediment might be eroded in regions of excessive agriculture (Galy et al., 2007; Garzanti et al., 2011), there might also be regions of sediment storage and river accretion. Wasson (2003) estimated the fraction of Ganges sediment discharge that was eroded from floodplain regions to be $< 10\,\%$. As an upper estimate, we assume these $10\,\%$ of Ganges sediment to be eroded directly within the floodplain section that yields the strongest GRACE gravity reduction (part of the Ganges catchment in $76\,°$E to $79\,°$E and $28\,°$N to $30\,°$N). For this area, the sediment loss would represent a mass loss of roughly $0.9\,\mathrm{kg\,m^{-2}\,yr^{-1}}$ and would explain at most $2\,\%$ of the observed TWS decrease in this region ($5.4\,\mathrm{cm\,yr^{-1}}$). Most likely, floodplain sediment would be eroded more homogeneously from the catchment, reducing the impact to less than $1\,\%$ of the observed gravity decrease. Thus, despite high sediment discharge in by Indian rivers, the impact of sediment mass loss on TWS trends in the floodplains is comparatively small.

### 3.5 Impact of the Himalaya uplift on geodetic observations of trends in terrestrial water storage

Sediment discharge is not the only process that impacts TWS trends from satellite gravimetry. One other process significant in the Himalayan study area is mountain orogeny. The Indian and Eurasion continental plates collide at a speed of about $50\,\mathrm{mm\,yr^{-1}}$ (Larson et al., 1999). This causes an uplift of the Himalayan mountain range (Bisht et al., 2021) and consequentially a mass increase within this collision region. Similar to the sediment transport by rivers, such tectonic processes have so far been considered too small to be observed via satellite gravimetry (Mikhailov et al., 2004). However, like the signal of sediment transport, this gravity change becomes relevant when studying trends over long time periods.

While the tectonic impact on satellite gravimetry is not the focus of our study, it is relevant in order to contextualize and interpret our study as well as for potential future application of our study results. Since the Indian plate moves below the Eurasian plate, the tectonic uplift is present in the Himalayan mountain ranges and in the Tibetean Plateau but not in the Indian floodplains (Li et al., 2020). We derived an estimate of the associated mass increase based on published uplift data (Xu et al., 2000; Fu and Freymueller, 2012; Bisht et al., 2021). For the Ganges and Brahmaputra mountain ranges, we find mass increases of $(0.8 \pm 1.1) \cdot 10^{12}\,\mathrm{kg\,yr^{-1}}$ and $(1.1 \pm 1.2) \cdot 10^{12}\,\mathrm{kg\,yr^{-1}}$, respectively. Details can be found in the supplemental material.

This mass increase caused by orogenic uplift in the Himalayan mountains is in the same order of magnitude as the mass reduction by the sediment transport in rivers. While both processes are present in the mountain ranges, uplift effects the full area and sediment erosion is the strongest along the river paths. However, at the current satellite resolution it is not possible to separate the two processes. Thus, the gravimetric impact of tectonic processes should be studied further and needs to be combined with the impact of sediment transport before attempting a correction of TWS trends from satellite gravimetry along tectonically active mountain ranges.

## 4 Conclusions

Our study shows the impact of sediment erosion on gravimetric estimates of TWS loss within main river catchments on the Indian subcontinent. Sediment erosion within the combined Ganges, Brahmaputra, Meghna, and Indus catchments yield an average mass loss of $(0.5\pm0.2)\,\mathrm{kg\,m^{-2}\,yr^{-1}}$ which potentially causes $4\,\%$ of the observed gravity decrease currently attributed to groundwater loss. Exclusion of the Indus catchment increases the sediment impact to approximately $5\,\%$.

Comparison of the sediment mass loss for individual river catchments yields the highest impact for the Brahmaputra catchment. There, sediment mass loss is $(1.1\pm0.4)\,\mathrm{kg\,m^{-2}\,yr^{-1}}$, corresponding to almost $8\,\%$ of observed gravity decrease within this catchment. In the Ganges catchment, sediment transport represents $3.3\,\%$ of the gravity decrease, while for the Meghna and Indus catchments its $2.2\,\%$ and $1.3\,\%$, respectively.

Mountain regions are especially prone to erosion. Thus, the impact of sediment mass loss on satellite gravimetry is especially important for mountain ranges. Over the whole Ganges and Brahmaputra mountain range, we find sediment mass loss of $(2.2\pm1.0)\,\mathrm{kg\,m^{-2}\,yr^{-1}}$ with average loss of $(3.4\pm1.8)\,\mathrm{kg\,m^{-2}\,yr^{-1}}$ in the Ganges mountains and $(1.7\pm0.7)\,\mathrm{kg\,m^{-2}\,yr^{-1}}$ in the Brahmaputra mountains. This represents $22\,\%$ and $10\,\%$ of the observed gravity decrease in the Ganges and Brahmaputra mountains, respectively. Inspection of previously stated erosion hotspots indicates that the sediment loss could potentially explain up to $77\,\%$ of the gravity decrease in selected mountain regions. However, investigation of the gravity increase caused by mountain orogeny yields data in the same order of magnitude as the gravity decrease by sediment discharge. Both processes are present mainly in the catchment mountain fractions, and at the current satellite resolution, it is not possible to separate the two processes. Thus, further studies of spatial distributions in sediment erosion and mountain orogeny are needed to better constrain their combined impact on satellite gravimetry over tectonically active areas.

In the river floodplains, where gravimetric measurements show the strongest decrease, the sediment impact is much smaller than over the mountains. The strongest gravity decrease is observed in north-west India with a reduction of up to $5.8\,\mathrm{cm}$ of EWH per year. In this area, we find the sediment impact to be at most $2\,\%$ with less than $1\,\%$ over the whole floodplain area.

*Author contributions.* AK and TW developed the concept of the study. AK performed the analysis and led the writing of the paper. MW and JM contributed to the interpretation of geodetic data. TR helped interpret the sediment data. HB, JN and CL contributed to the general data interpretation. All authors discussed results and commented on the manuscript.

*Competing interests.* The authors declare no competing interests.

*Acknowledgements.* We gratefully acknowledge the support of the Norddeutscher Wissenschaftspreis (North German Science Prize), which provided funding that enabled our studies in the interdisciplinary realms of Geodesy and Climate Research. Additionally, we express our gratitude to Maxime Mouyen and an anonymous referee for their insightful and constructive feedback, which enhanced the quality of this paper.

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
