# Peer review of "Sediment transport in South Asian rivers high enough to impact satellite gravimetry"

_Hydrology and Earth System Sciences, 2023_

## Referee Comment (RC1)

**Review of "Sediment transport in Indian rivers high enough to impact satellite gravimetry" by Alexandra Klemme et al.**

The manuscript evaluates the potential impact of erosion on gravity changes measured by GRACE and GRACE Follow-on with the aim to better decipher mass redistribution due to hydrological processes from mass redistribution due to erosion. The study focuses on the Himalayan region and the catchments of the Ganges, Brahmaputra, Meghna and Indus rivers, where sediment discharge rates are amongst the highest on Earth and where hydrological processes are also active and responsible for most of the gravity variations. The work is centered on compiling all available information regarding sediment discharge rates in the area, converting them into "GRACE-like" signals (commonly expressed in equivalent water height) and comparing such effects to the actual GRACE observation. The authors eventually derive erosion-induced gravity effects that should be accounted for when interpreting GRACE time series in the region, especially when one wants to properly quantify groundwater depletion and other processes that redistribute water.

The topic is interesting and within the scope of Hydrology and Earth System Sciences. The paper is concise and well-written, the compilation of sediment discharge data represents significant work, but the method should be improved and the leading hypotheses of the work should be clarified because one could argue that no net sediment mass loss may exist at all in the studied area. These two aspects, method and hypotheses, are major weaknesses that the authors must address. I presently do not recommend the paper for publication but would be happy to review an improved version of the manuscript.

I will develop my comments below.

**Major comments**

**About the working hypothesis:**

Since the Himalayan orogenesis is still active, can one really expect to observe a net sediment loss in the area? The paper should clearly state whether the time scales considered in the study (GRACE times scales, i.e. 20 years) is appropriate to assume that only erosion processes are at work in the area, without any mass gain due to the orogenic process itself. All the processes are presented as continuous in time and studied in linear trends (gravity changes, sediment discharge). So in the same logic, there should exist in this region a continuous trend of mass increase sustained by the collision between the Indian and Eurasian plates. The study tacitly assumes that there is no equilibrium orogeny/erosion over the study period. That is a very strong hypothesis that must be extensively argued.

**About the method:**

P3 L69-70: I don't understand the separation in catchment fractions. I agree that it is important to spatialize the quantifications, especially given the rather high-resolution of the GRACE solutions you use, but the distinction agricultural vs mountain is not obvious and not explained in the text. Also what about the remaining area, are they just by-pass areas where no erosion/sedimentation occurs? The study would benefit from more elaborated spatialization methods, e.g. using the concept of topographic index, which quantifies if an area is prone to accumulate or loose materials (originally developed water). See eg

https://topotoolbox.wordpress.com/ and DEM such as SRTM to infer such a parameter and proceed with a potentially refined spatialisation of the study.

It would be eventually very interesting to assess the spatial distribution of sediments mass variations over the entire catchment, converted to the same spatial scale and units as the GRACE data. This would help to highlight the specific regions where effects of erosion must be accounted for when studying GRACE time series.

The modeling of erosion effect on the GRACE signal should be improved by accounting for seasonality. At page 7 line 121, the seasonality of sediment discharge is dismissed from the analysis. I found this decision a bit drastic, especially in light of the efforts you showed in the appendix to describe this seasonality. It is possible to redistribute the sediment discharge over the monsoon periods only, and proportionally to the change of EWH. Then how are the GRACE rates altered?

Still on this seasonality aspect, several hydrological models exist are often compared to GRACE. They often miss interannual features (1) but are usually performing well at seasonal time scales (2). I think these seasonal hydrological aspects must be investigated, because they will interfere significantly with sediment discharge. For instance:

- How are linear trends altered by such hydrological corrections?
- What are the seasonal residuals after such models have been removed and what would be the relative part of the sediment mass variation in these residuals?
- Other questions may arise depending on how the residuals look.

(1) https://doi.org/10.1073/pnas.1704665115

(2) https://doi.org/10.5194/hess-21-821-2017

**Minor comments**

Table 1: the catchments areas in the first row don't add up in GBM and Total

P10 L174 and a few other places: Use "decrease/negative linear trend" or any other terms that is more explicit than "anomaly".

Following my first main comment about the time scale of the study: Ideally, the time needed for the eroded materials to travel from their sources to outside their catchment basin should also be taken into account, but this may go beyond the scope of the study.

---

## Author Comment (AC1)

**Reply on comments by Referee #1**

Referee:
The manuscript evaluates the potential impact of erosion on gravity changes measured by GRACE and GRACE Follow-on with the aim to better decipher mass redistribution due to hydrological processes from mass redistribution due to erosion. The study focuses on the Himalayan region and the catchments of the Ganges, Brahmaputra, Meghna and Indus rivers, where sediment discharge rates are amongst the highest on Earth and where hydrological processes are also active and responsible for most of the gravity variations. The work is centered on compiling all available information regarding sediment discharge rates in the area, converting them into "GRACE-like" signals (commonly expressed in equivalent water height) and comparing such effects to the actual GRACE observation. The authors eventually derive erosion-induced gravity effects that should be accounted for when interpreting GRACE time series in the region, especially when one wants to properly quantify groundwater depletion and other processes that redistribute water.

The topic is interesting and within the scope of Hydrology and Earth System Sciences. The paper is concise and well-written, the compilation of sediment discharge data represents significant work, but the method should be improved and the leading hypotheses of the work should be clarified because one could argue that no net sediment mass loss may exist at all in the studied area. These two aspects, method and hypotheses, are major weaknesses that the authors must address. I presently do not recommend the paper for publication but would be happy to review an improved version of the manuscript.

I will develop my comments below.

Response:
We thank the reviewer Maxime Mouyen for their work on our manuscript. The study summary highlights an understanding of the topic and the goals of our manuscript. While they point out shortcomings regarding our methodology and the impact of orogenic processes in the Himalayas, we are pleased with the generally positive response to the study concept. We are happy to adapt our manuscript based on the suggestions. A detailed response is provided in the following.

Referee:
**Major comments**
*About the working hypothesis:*
Since the Himalayan orogenesis is still active, can one really expect to observe a net sediment loss in the area? The paper should clearly state whether the time scales considered in the study (GRACE times scales, i.e. 20 years) is appropriate to assume that only erosion processes are at work in the area, without any mass gain due to the orogenic process itself. All the processes are presented as continuous in time and studied in linear trends (gravity changes, sediment discharge). So in the same logic, there should exist in this region a continuous trend of mass increase sustained by the collision between the Indian and Eurasian plates. The study tacitly assumes that there is no equilibrium orogeny/erosion over the study period. That is a very strong hypothesis that must be extensively argued.

Response:

The referee is correct in pointing out the gravity impact of orogenic processes in the Himalayas. We have included a discussion of this to out manuscript. In the Introduction, we state: *»Additional processes to consider in long-term gravimetric data are plate tectonics. The Himalaya Mountain range experiences uplift due to the tectonic collision between the Indian and the Eurasian continental plates. The gravimetric impact of this process is not the focus of this study. Yet, knowledge of such additional tectonic process is essential to contextualize the resulting sediment impact, as the increase in mass due to this Himalayan Mountain uplift could counteract part of the mass loss due to sediment erosion and discharge.«*

We derived an estimate for the impact of Himalayan orogeny on satellite gravimetry. Details of this are included in a new section in the supplement. In the main manuscript, we have included a new discussion section 3.5 titled: *»Impact of the Himalaya uplift on geodetic observations of trends in terrestrial water storage«*. This section includes the following paragraphs:

*»Sediment discharge is not the only process that impacts TWS trends from satellite gravimetry. One other process significant in the Himalayan study area is mountain orogeny. The Indian and Eurasian continental plates collide at a speed of about 50 mm yr$^{-1}$ (Larson et al., 1999). This causes an uplift of the Himalayan Mountain range (Bisht et al., 2021) and consequentially a mass increase within this collision region. Like the sediment transport by rivers, such tectonic processes have so far been considered too small to be observed via satellite gravimetry (Mikhailov et al., 2004). However, like the signal of sediment transport, this gravity change becomes relevant when studying trends over long time periods.*
*While the tectonic impact on satellite gravimetry is not the focus of our study, it is relevant in order to contextualize and interpret our study as well as for potential future application of our study results. Since the Indian plate moves below the Eurasian plate, the tectonic uplift is present in the Himalayan Mountain ranges and in the Tibetan Plateau but not in the Indian floodplains (Li et al., 2020). We derived an estimate of the associated mass increase based on published uplift data (Xu et al., 2000; Fu et al., 2012; Bisht et al., 2021). For the Ganges and Brahmaputra mountain ranges, we find mass increases of $(0.8 \pm 1.1) \cdot 10^{12}$ kg yr$^{-1}$ and $(1.1 \pm 1.2) \cdot 10^{12}$ kg yr$^{-1}$, respectively. Details can be found in the supplemental material.*
*This mass increase caused by orogenic uplift in the Himalayan mountains is in the same order of magnitude as the mass reduction by the sediment transport in rivers. While both processes are present in the mountain ranges, uplift effects the full area and sediment erosion is the strongest along the river paths. However, at the current satellite resolution it is not possible to separate the two processes. Thus, the gravimetric impact of tectonic processes should be studied further and needs to be combined with the impact of sediment transport before attempting a correction of TWS trends from satellite gravimetry along tectonically active mountain ranges.«*

Finally, in the study conclusion, we follow the paragraph on sediment mass loss in the mountains by the statement that *»investigation of the gravity increase caused by mountain orogeny yields data in the same order of magnitude as the gravity decrease by sediment discharge. Both processes are present mainly in the catchment mountain fractions, and at the current satellite resolution, it is not possible to separate the two processes. Thus, further*

*studies of spatial distributions in sediment erosion and mountain orogeny are needed to better constrain their combined impact on satellite gravimetry over tectonically active areas.«*

**Referee:**
*About the method:*

P3 L69-70: I don't understand the separation in catchment fractions. I agree that it is important to spatialize the quantifications, especially given the rather high-resolution of the GRACE solutions you use, but the distinction agricultural vs mountain is not obvious and not explained in the text.

**Response:**

Here, our methodology seems to have been unclearly stated. We have not separated between GRACE data over mountain and agricultural regions. These regions were identified to derive fractions of mountain and agricultural areas within specific catchments. These catchment fractions help us to better interpret GRACE signals in the catchments. To avoid confusion in the manuscript, we remove such regions from Figure 1, and in the main manuscript only refer to the derived catchment fractions stated in Table 1. The figure including mountain and agricultural regions is shifted to the supplement.

**Referee:**

Also what about the remaining area, are they just by-pass areas where no erosion/sedimentation occurs? The study would benefit from more elaborated spatialization methods, e.g. using the concept of topographic index, which quantifies if an area is prone to accumulate or loose materials (originally developed water). See eg https://topotoolbox.wordpress.com/ and DEM such as SRTM to infer such a parameter and proceed with a potentially refined spatialisation of the study.

It would be eventually very interesting to assess the spatial distribution of sediments mass variations over the entire catchment, converted to the same spatial scale and units as the GRACE data. This would help to highlight the specific regions where effects of erosion must be accounted for when studying GRACE time series.

**Response:**

According to the referee's suggestion, we investigated the topographic impact on erosion potential and considered a spatial distribution of erosion according to soil loss derived from the Revised Universal Soil Loss Equation (RUSLE). However, these results do not agree with published estimates of sediment origin distributions based on field studies in the area. We decided to continue to base our results on those local studies. For comparison, the spatial distribution of soil erosion based on RUSLE is included in the revised supplement. In the main manuscript, we state that *»Due to data scarcity, it is difficult to assess spatially resolved data for sediment induced gravity changes in the Indian subcontinent. In the following, we separate between sediment eroded from specific mountain regions based on published literature (Wasson et al., 2003; Galy et al., 2007; Faisal et al., 2022). Additionally, a discussion of spatially resolved sediment loss based on soil loss data from the Revised Universal Soil Loss Equation (RUSLE, Borrelli et al., 2017) is included in the Supplemental Material.«*

Referee:

The modeling of erosion effect on the GRACE signal should be improved by accounting for seasonality. At page 7 line 121, the seasonality of sediment discharge is dismissed from the analysis. I found this decision a bit drastic, especially in light of the efforts you showed in the appendix to describe this seasonality. It is possible to redistribute the sediment discharge over the monsoon periods only, and proportionally to the change of EWH. Then how are the GRACE rates altered?

Response:

This seems to have been unclearly communicated in our study. We have included the sediment seasonality in our data correction. However, the seasonality is about three orders of magnitude smaller than the observed GRACE seasonality, which means that its essentially lost within the GRACE uncertainties. Therefore, we dismiss the significance of such seasonality for our study.

To clarify this, we have shifted the discussion of seasonality to the main text of our manuscript and included a figure showing the seasonality of sediment loss in units of EWH. Additionally, we explicitly state that »*the sediment mass loss in units of EWH show values that are by three orders of magnitude smaller than the seasonality observed in GRACE data. This monthly sediment impact is within the uncertainty of monthly gravimetry data and will not considerably impact this study's analysis. While seasonality is included in the following data, we will from here on focus on linear trends in both water and sediment loss.*«

Referee:

Still on this seasonality aspect, several hydrological models exist are often compared to GRACE. They often miss interannual features (1) but are usually performing well at seasonal time scales (2). I think these seasonal hydrological aspects must be investigated, because they will interfere significantly with sediment discharge. For instance:
- How are linear trends altered by such hydrological corrections?
- What are the seasonal residuals after such models have been removed and what would be the relative part of the sediment mass variation in these residuals?
Other questions may arise depending on how the residuals look.
(1) https://doi.org/10.1073/pnas.1704665115
(2) https://doi.org/10.5194/hess-21-821-2017

Response:

We appreciate the suggestion to look deeper into the discharge seasonality. However, as mentioned before, the impact of seasonality in sediment is by three orders of magnitude smaller than the GRACE seasonality and well within the GRACE data uncertainty. Thus, we consider the analysis of linear changes within the catchments to be sufficient at the current availability of data.

Referee:

**Minor comments**
Table 1: the catchments areas in the first row don't add up in GBM and Total

Response:

This inconsistency was corrected.

Referee:
P10 L174 and a few other places: Use "decrease/negative linear trend" or any other terms that is more explicit than "anomaly".

Response:
This has been changed accordingly.

Referee:
Following my first main comment about the time scale of the study: Ideally, the time needed for the eroded materials to travel from their sources to outside their catchment basin should also be taken into account, but this may go beyond the scope of the study.

Response:
This would ideally be the case. Yet, considering the scarcity of sediment data and the extensive re-distribution within the catchments, we consider it outside the capacity of our study. We rely on sediment data close to the river delta since we know this sediment left the catchment. However, at the current state, we cannot distinguish whether sediment has been directly transported from the mountains or extensively redistributed in the floodplains.

---

## Author Comment (AC2)

**Reply on comments by Referee #2**

Referee:

The paper "Sediment transport in Indian rivers high enough to impact satellite gravimetry" by Klemme et al. examines how sediment transport can affect trends in gravity fields observed by the GRACE satellites in several river basins in India. This is an important study as accounting for sediment transport can directly affect how we interpret GRACE derived terrestrial water storage changes. The paper is concise and well written. But the manuscript is too focused on impacts on trends. I think the manuscript can be improved substantially if the authors can add some analyses on sediment data.

Response:

We thank the reviewer for their work on our manuscript and appreciate their constructive feedback. We are happy to incorporate their suggestions and improve the manuscript. Our detailed response can be found in the following.

Referee:

My major comments are:

As the most important data for this study, sediment data are not well analyzed and presented, making it difficult to assess the quality of the research. At minimum, there should be an analysis on seasonal and interannual variability of sediment data and their correlations with precipitation and GRACE EWH in each basin. There is a figure on seasonal variation of sediments but it is buried in the Supplementary file.

In addition, an analysis on how temporal variability of sediments varies from one basin to another would be helpful to understand their climate and environmental controls. If sediments eventually end up at the Bay of Bengal, do sediment data collected at the Bay of Bengal show higher seasonal maximum and lagged correlations with sediments at each basin? These analyses will establish the basis for the need to consider the impact of sediment loss on gravity changes. To accompany these analyses, I suggest a paneled figure that shows time series of sediments, GRACE EWH and precipitation data for each of the basins and for all basin average.

Response:

We agree that the presentation of sediment data within our study is important. Unfortunately, the current data availability does not allow for a detailed discussion of seasonal or interannual variability within the catchments. We have shifted the discussion of data seasonality to the main manuscript and included statements on the differences between individual catchments. We need to highlight however, that the seasonality in sediment discharge is based on seasonality in the river's water discharge rather than sediment measurements.

Referee:

Seasonality (i.e., monthly mean) needs to be removed from the GRACE EWH time series before computing any trends. Strong seasonality as in GRACE data may affect computing long-term trends. Related to this issue, seasonal cycles should be removed from Figs.4-6 to make the differences in trends more discernable. Seasonal variations can be shown in the figure suggested above.

Response:

Based on the reviewer's comment, we performed a more detailed trend analysis to decipher the impact of seasonality on the linear trend. For this, we utilized a dynamic linear model, allowing for variable seasonality and interannual trends. We then derived a median average trend from the best twelve model results. All derived trends, within their uncertainties, agree with the linear trends used in the study. Relative differences are within 5 % and absolute differences are smaller than 0.1 cm yr$^{-1}$. We proceed to use the linear trends in the study, but include results from this model analysis in the supplement. In the main manuscript, we state: *»Linear least-squares optimizations of the generated monthly time-series yield the local TWS trends. [...] A more detailed trend analysis is included in the supplemental material.«*

For the correction figures, we have decided to move the figure for the full study area to the supplement and instead replace it by a trend comparison as the reviewer suggested below. We hope, this will help to convey the information the reviewer found hard to discern in the initial figures. For the other two correction figures, we decided to leave the seasonality in the data, as it helps convey an understanding of the dimension in change of the trend compared to the natural TWS seasonality.

Referee:

At the end of reading section 3.3, those numbers no longer register with me. Since all the numbers are provided in tables, there is no need to state them in the text. Instead, the manuscript should highlight the largest impacts or patterns of impacts that may be interesting to readers. A scatter plot showing TWS trends without correction for sediments vs those with the corrections would be useful for identifying patterns and for accompanying the manuscript.

Response:

We have limited the numbers provided in the text to the most essential ones and include a comparison plot of the TWS data as suggested.

Referee:

Given the coarse scale of GRACE data and the lack of detailed sediment data, section 3.3.4. is flimsy. If included, the authors need to show their calculations and provide justification for assumptions made in Lines 184-187 and line 188-192.

Response:

We understand where the reviewer is coming from. This section is included to illustrate the impact of sediment discharge on the floodplains, where the main groundwater depletion is taking place. Given the scarcity in data, it can only be a rough estimate. We have re-phrased the section. It is now titled *»Impact within floodplain regions«* and we state that *»To estimate the impact of sediment discharge on gravity data of groundwater depletion, we are interested in erosion within the Indo-Gangetic floodplain, where the strongest gravity decrease is observed. Generally, the estimation of the sediment impact in river lowlands and floodplains is more complicated than in mountain regions due to sedimentary redistribution within the catchments. While some sediment might be eroded in regions of excessive agriculture (Galy el at., 2007; Garzanti et al., 2011), there might also be regions of sediment storage and river accretion. Wasson et al. (2003) estimated the fraction of Ganges sediment discharge that was eroded from floodplain regions to be < 10 %. As an upper estimate, we assume these 10 % of*

*Ganges sediment to be eroded directly within the floodplain section that yields the strongest GRACE gravity reduction (part of the Ganges catchment in 76°E to 79°E and 28°N to 30°N). For this area, the sediment loss would represent a mass loss of roughly 0.9 kg m$^{-2}$ yr$^{-1}$ and would explain at most 2 % of the observed TWS decrease in this region (5.4 cm yr$^{-1}$). Most likely, floodplain sediment is eroded more homogeneously from the catchment, reducing the impact to less than 1 % of the observed gravity decrease. Thus, despite high sediment discharge in by Indian rivers, the impact of sediment mass loss on TWS trends in the floodplains is comparatively small.«*

Referee:

Minor comment:

Fig.1.  The white color for high elevation is invisible.

Response:

We changed the colour scheme accordingly. The Figure has been moved to the Supplement.

Referee:

Line 56: the clause after whereat needs to be revised for clarity.

Response:

We deleted the clause, since this information is now conveyed in the new seasonality section.

Referee:

Line 120: EWH increase and EWH decrease may be replaced by "high EWH values" and "low EWH values", respectively.

Response:

We decided to leave this as is, since it is in fact the time of increasing EWH values (positive slope) and decreasing EWH values (negative slope) rather than high and low values that is referred to.

Referee:

Figs.4&5 contain references to σ-environment which is not explained anywhere else in the manuscript.

Response:

In the revised manuscript we specify that the σ-environment refers to the standard deviation stated in Table 3.

Referee:

Line 10-14: The sentence is too long and difficult to understand.  Please revise.
Line 20: e.g. is not correctly used here.  Replace "on e.g." with "such as"
Line 23: explain it -> explained it
Line 36: annual -> interannual?
Line 138: please delete ", with".
Line 150: reduction in GRACE EWH ->decreasing trend in GRACE EWH?

Response:

These points were changed accordingly.